# Auxological and Metabolic Parameters of Children Undergoing the Gonadotropin-Releasing Hormone Stimulation Test: Correlations with the Final Diagnosis of Central Precocious Puberty in a Single-Center Study

**DOI:** 10.3390/biomedicines11061678

**Published:** 2023-06-09

**Authors:** Clelia Cipolla, Giorgio Sodero, Lucia Celeste Pane, Francesco Mariani, Lorenzo Di Sarno, Donato Rigante, Marcello Candelli

**Affiliations:** 1Department of Life Sciences and Public Health, Fondazione Policlinico Universitario A. Gemelli IRCCS, 00168 Rome, Italy; cipolla.clelia@gmail.com (C.C.); giorgio.sodero@hotmail.it (G.S.);; 2Università Cattolica Sacro Cuore, 00168 Rome, Italy; 3Department of Emergency Medicine, Fondazione Policlinico Universitario A. Gemelli IRCCS, 00168 Rome, Italy; mcandelli@gmail.com

**Keywords:** central precocious puberty, GnRH test, pediatric endocrinology, personalized medicine

## Abstract

Background—Central precocious puberty (CPP) is characterized by clinical, biochemical, and radiological features similar to those of normal puberty, but CPP occurs before the age of eight in girls and before the age of nine in boys, subsequently leading to a reduction in the final body height in adulthood due to premature fusion of growth plates. The diagnosis of CPP is confirmed with a gonadotropin-releasing hormone (GnRH) stimulation test, which can lead to different interpretations because the diagnostic peak levels of luteinizing hormone (LH) can vary. Patients and methods—This was a single-center, retrospective observational study investigating the possible correlation between gonadotropin peaks on the GnRH test and auxological, metabolic, and radiological parameters of patients evaluated for CPP. We collected and analyzed data from the medical records of children with suspected CPP over a period from January 2019 to July 2022 who underwent a GnRH test at the Fondazione Policlinico Universitario Agostino Gemelli in Rome, Italy. Results—Our correlation analysis revealed no statistically significant differences in any auxological and radiological parameters. Among laboratory parameters, baseline levels of LH, follicle-stimulating hormone, sex hormone-binding globulin, and 17-beta estradiol were higher in children with a definitive diagnosis of CPP than in those with a negative GnRH test. In particular, the levels of LH at baseline and after the GnRH test were statistically significant in the group of CPP patients, consistent with the interpretation of the test. In the multivariate analysis, using a cut-off value of 4.1 IU/L, LH peaks showed both very high sensitivity (94%) and very high specificity (95%); all other variables showed high specificity (90%) but unsatisfactory sensitivity. Conclusion—Basal hormone dosages and, especially, basal levels of LH should be considered before performing a GnRH test as they might anticipate the final diagnosis of CPP.

## 1. Introduction

Central precocious puberty (CPP) is a pathological condition characterized by clinical, biochemical, and radiological features that resemble ‘normal’ puberty, which otherwise occurs before the age of eight in girls and before the age of nine in boys, due to premature activation of the hypothalamic–pituitary–gonadal axis [1]. Gonadotropin-releasing hormone (GnRH) produced by the hypothalamus stimulates the cells in the anterior pituitary gland to release luteinizing hormone (LH) and follicle-stimulating hormone (FSH), which trigger the final production of steroid hormones in the ovaries or testes. Premature and inappropriate development of secondary sexual characteristics is usually associated with an accelerated growth rate leading to progressive and premature closure of the growth plates [2] and a consequent reduction in the final adult height.

In girls, who represent the majority of children with PPC, the first clinical symptom is breast development (also called thelarche), followed by pubic hair growth (pubarche), statural growth spurt, and menarche, whereas, in boys, the first clinical feature of CPP is genital growth with insufficient testicular volume for the child’s age [3]. Moreover, CPP is largely idiopathic in girls, while in boys it may be associated with underlying diseases such as central nervous system tumors, although genetic forms caused by mutations in the kisspeptin pathway that increase the pulsatility of GnRH secretion have been reported [3,4].

The diagnosis of CPP is suspected on a clinical basis by the pediatrician or by the pediatric endocrinologist and confirmed with a GnRH stimulation test [4], which is the most important tool for diagnosis. Although, the test is subject to variations in both modalities of performance and interpretation of results, as the diagnostic LH peak can vary from 3.3 IU/L to 8-10 IU/L depending on the guidelines used [1,4,5].

The aim of the present study was to evaluate the potential correlation between GnRH stimulation test results and auxological, metabolic, and radiological parameters in a cohort of children evaluated for suspected CPP and highlight the differences in these parameters between patients for whom the final diagnosis of CPP was made and those for whom this diagnosis was excluded.

## 2. Patients and Methods

### 2.1. Study Characteristics

This is a single-center, retrospective observational study investigating any possible correlations between LH and FSH peaks during the GnRH test and auxological, metabolic, and radiological parameters of children being evaluated for CPP. Ethics committee approval was not obtained because the General Authorization to Process Personal Data for Scientific Research Purposes (Authorization No. 9/2014) states that retrospective archival studies using ID codes that prevent direct tracing of data to the subject do not require formal ethics approval [6]. The parents of the patients studied were informed about the purpose of this study and signed an informed consent form for consenting access to children’s medical records and for processing personal data.

### 2.2. Patient Selection

We collected and analyzed data from the medical records of all children who underwent a GnRH stimulation test in our hospital (Fondazione Policlinico Universitario Agostino Gemelli IRCCS) for the clinical suspicion of CPP during the period from January 2019 to July 2022. The initial screening allowed a pre-selection of 52 children evaluated in the pediatric endocrinology outpatient clinic for suspected CPP. After this initial selection, clinical and laboratory data were collected from patients who subsequently underwent GnRH testing, for a total of 37 children (2 males and 35 females). All remaining patients were lost to follow-up despite clinical suspicion of CPP (they refused to be tested or decided to be followed up at other hospitals).

Medical records were retrospectively analyzed, and information on auxological parameters (weight, height, body mass index (BMI), adrenarche, pubarche, menarche), metabolic parameters (laboratory tests performed before the stimulus test, i.e., thyroid hormones, blood glucose, insulin, complete blood cell count with leukocyte count, transaminases (aspartate aminotransferase, GOT), lipid profile, FSH, LH, progesterone, 17-beta estradiol (17BE2), sex hormone binding globulin (SHBG), prolactin), radiological parameters (bone age, gynecological ultrasonography, brain magnetic resonance imaging), and those related to the stimulus test (LH and FSH results, LH/FSH ratio) were methodically collected. The data were collected in an electronic database used for statistical analysis. The characteristics of this cohort of children are summarized in Table 1.

The auxological parameters were studied after calculating the reference percentiles and standard deviation (SD) for the age and sex of each patient to obtain homogeneous and comparable data. Thyroid and lipid tests were considered abnormal if they were above the 90^th^ percentile for age and sex; abnormal blood counts were assessed by comparing hemoglobin and hematocrit values with the corresponding percentiles; hyperglycemia was determined if fasting glucose was equal to or higher than 126 mg/dl without signs, symptoms, or other criteria for diabetes; GOT, FSH, LH, and prolactin levels were considered elevated if they were above the standard normal values in prepubertal children. Ultrasound parameters were considered abnormal if they were above those recommended in our national guidelines for any prepubertal child (uterus length ≥ 3.6 cm, uterus anteroposterior diameter > 1 cm, uterine corpus/cervix ratio > 1, ovarian volume ≥ 2 mL) [4].

### 2.3. Characteristics of the GnRH Stimulation Test

The diagnosis of CPP is not fully standardized and requires the integration of clinical, laboratory, and radiological data [7]. However, all guidelines for CPP diagnosis identify the GnRH test as a crucial tool. This test is performed using the intravenous infusion of up to 100 mcg of GnRH followed by a serial sampling of LH and FSH at 0, 30, 60, 90, and 120 min [4]. CPP can be confirmed if LH shows a peak higher than 3.3 or 5.0 IU/L (depending on the guidelines used), also taking into account the LH/FSH ratio after the stimulus, which, if higher than 0.6 or 1 (based on the guidelines used), suggests CPP. In our center, we regularly use a cut-off value of 3.3 IU/L for the evaluation of the LH peak. At lower values, the patient is followed up in the endocrinology clinic to assess the growth rate, the possible development of secondary sexual characteristics, and the need for further testing to rule out other causes of early puberty. Once diagnosed, children with CPP begin specific therapy with GnRH analogs administered intramuscularly at a dosage of 3.75 mg every 3–4 weeks until they reach the correct age for the onset of puberty [1].

### 2.4. Statistical Analysis

Statistical analyses were performed using IBM SPSS Statistics 25.0 software (IBM Corporation, Armonk, NY, USA). For continuous variables, the Kolmogorov–Smirnov test was used to assess whether the distribution was normal or not. Categorical variables were expressed as count and percentage. Continuous variables with normal distribution were expressed as the mean with SD, and data with a skewed distribution were expressed as the median with an interquartile range (IQR 25–75%). Statistical comparisons between groups were obtained using the chi-squared test or Fisher’s exact test for categorical variables and the Mann–Whitney U-test or *t*-test for continuous variables. A *p*-value < 0.05 was considered statistically significant.

We also performed multivariate analysis (linear logistic regression) including all variables that had a *p*-value < 0.1 in the univariate analysis, after adjusting for confounding factors such as sex and age. In addition, receiver operating characteristic (ROC) curves were constructed for all variables that were found to be significantly associated with CCP in the multivariate analysis, and their diagnostic accuracy was assessed by calculating the area under the curve. Finally, we calculated the best cut-off value for each variable with its specificity and sensitivity for the diagnosis of CCP using the Youden index.

## 3. Results

### 3.1. Main Results

In our cohort of 37 patients who underwent GnRH testing, we found 18 cases of confirmed CPP (51.35%), while 19 cases were negative (48.65%). Overall, 6 out of 37 children (16.2%) had a familiar history of CPP, and 4 patients (66.6%) out of these had a subsequent diagnosis of CPP. All CPP patients were females (100%). Sixteen cases of CPP (88.9%) were interpreted as idiopathic, whereas two patients (11.1%) were found to have an organic cause: two central nervous system tumors (noninvasive low-grade glioma), in one case, associated with empty sella syndrome. None of the patients in our study received any pharmacological therapies before the GnRH test.

We compared the two groups and investigated the possible differences between children with and without CPP in pre-test auxological, metabolic, and radiological parameters. Our correlation analysis revealed no statistically significant differences for auxological or radiological parameters. Among laboratory parameters, baseline levels of hormones (LH, FSH, SHBG, and 17BE2) were higher in children with a definitive diagnosis of CPP than in those with a negative test (*p*-values are shown below). With regards to prolactin and other laboratory parameters (not necessarily related to the hypothalamic–pituitary axis activation), our analysis revealed no statistically significant differences. However, the difference between the two groups in the peak levels of LH and FSH during the GnRH test was statistically significant; this result is consistent with the test interpretation and diagnosis of CPP, which requires a peak level of LH higher than 3.3 IU/L. Further details are summarized in Table 2.

### 3.2. Multivariate Analysis

The results of the linear logistic regression after correction for confounding factors (sex and age) are shown in Table 3. Among the factors included in the analysis, LH, 17BE2, LH peaks, FSH peaks, and LH/FSH ratios were independently and significantly associated with CPP. In contrast, no significant association was found between lipid profile, FSH, SHBG, fT4, and CPP.

We then generated ROC curves for all variables that were significantly associated with CPP in the multivariate analysis. We also calculated diagnostic accuracy and evaluated the best cut-off for sensitivity and specificity for the diagnosis of CPP (Figure 1 and Table 4). The variable with the highest accuracy (0.988) was the LH peak: this result is in line with our guidelines, which support diagnosing CPP in the case of increased LH response to the GnRH test. Using the cut-off value of 4.1 IU/L, the LH peaks showed both very high sensitivity (94%) and specificity (95%). All other variables showed high specificity (90%) but unsatisfactory sensitivity, except for the LH /FSH ratio, whose sensitivity and specificity were lower (as shown in Table 4).

## 4. Discussion

The diagnosis of CPP is challenging and requires a multidisciplinary approach [4]. Once a CPP diagnosis is established, GnRH therapy is initiated to halt the progression of pubertal development. GnRH administration is generally safe, although it can lead to acute side effects (pain, itching, and sweating at the injection site) and, in a few cases, even long-term side effects such as pseudotumor cerebri or changes in bone mineral density [8]. One of the most common side effects is vaginal bleeding, usually spotting, which occurs only after the first administrations [9]. In contrast, anaphylactic reactions are rare [10].

In our center, a progressive increase in new cases of CPP after the SARS-CoV-2 pandemic has been observed in parallel with national incidence data on CPP, showing an acceleration of pubertal development in children compared with the period before COVID-19 [11]. Possible explanations for this trend include the psychological impact of the pandemic, an increase in weight and BMI during the lockdown, a deterioration in dietary habits, and decreased physical activity. However, in the present study, we did not examine diet or eventual nutritional supplementation in our patients.

To date, the GnRH stimulation test is the gold standard for confirming CPP because it detects ‘early’ activation of the hypothalamic–pituitary axis [4,10]. However, the performance and interpretation of the results can be difficult due to the different cut-off values suggested by scientific societies. Most recommendations consider LH above 5 IU/L or LH/FSH ratio above 1 (or 0.6, depending on the guidelines used) after a GnRH test as the most appropriate seal for the diagnosis of precocious puberty in children [12], although Italian recommendations consider even a peak above 3.3 IU/L useful [4].

Multivariate analysis performed in our sample of children with suspected CPP showed that a cut-off value of 4.1 IU/L for the LH peak had excellent diagnostic accuracy compared with other cut-off values from other studies. This value is higher than the one used in our guidelines (3.3) but lower than in other international recommendations (5.8 or 10).

Further important issues are the total duration of the GnRH stimulation test and the need to take several blood samples to assess peak hormone levels. There are several studies in the medical literature that have attempted to change the text rules. For example, Yeh et al., studied 313 patients with suspected CPP who underwent 381 GnRH tests and demonstrated that a single evaluation of LH 30 min after the start of the stimulus was equivalent to performing the entire test. However, the authors used different cut-off values than we used (LH peak positive for levels > 10 IU/L), and lower elevations were not detected [13]. Similar results were also obtained by Kim et al., who studied 166 girls undergoing a GnRH test and showed that in 128 cases of CPP, 98.4% of blood samples taken after 45 min were sufficient for CPP diagnosis (LH cut-off > 5 IU/L). Moreover, the combination of samples taken after 30 and 45 min resulted in 100% of diagnoses, in agreement with the results from the full test [14].

The results of our study showed that basal hormonal parameters of children with suspected CPP may be correlated with the final result of the GnRH test, especially the increased basal LH level (*p* = 0.03). We also found a statistically significant correlation with other basal hormonal levels (FSH, SHBG, and 17BE2) that, although unable to diagnose CPP, may support the diagnosis if associated with elevated LH levels. The multivariate analysis confirmed the excellent sensitivity of most parameters, albeit with lower specificity. Therefore, the basal level of LH and 17BE2 as well as the level of FSH can predict the diagnosis of CPP but without replacing the performance of a GnRH test and a correct interpretation of the LH peak. The analysis of the LH/FSH ratio did not lead to satisfactory results in terms of the accuracy of CPP diagnosis: this is in line with the current diagnostic recommendations, which consider this parameter as secondary.

We also performed a similar analysis on children with short stature who underwent growth hormone (GH) stimulation tests. In this case, we found increased serum TSH levels and decreased IGF-1 levels in patients with GH deficiency compared to those with idiopathic short stature, and it became clear how both weight and BMI might influence post-test GH peaks [15]. A direct relationship between the BMI and GnRH test was also found in another retrospective study of 865 girls with CPP, which showed that a higher BMI, especially in obese patients, was associated with a lower LH response to stimulation with GnRH and a lower LH/FSH ratio [16]. These results were also confirmed by another study by Lee et al. [17], in which 981 girls with idiopathic CPP and an LH peak higher than 5 IU/L were analyzed. The authors stratified patients based on their Tanner stage and found that the negative correlation between BMI and stimulated LH was not confirmed in the most advanced pubertal stages (Tanner 4), whereas it occurred in the earliest stages (Tanner 2 and 3).

To date, there are no standardized percentiles for gynecologic measurements in Italian prepubertal children. During ultrasound examinations, we did not observe any differences between patients with and without CPP. In fact, gynecologic ultrasonography alone is not sufficient to diagnose CPP [4], although it might provide indirect information about gonadotropin secretion, such as an increase in ovarian volume or in uterine size. Our result can be firstly explained considering the small cohort size, but also using the required ultrasound features, as we regarded ‘abnormal’ a uterus length ≥ 3.6 cm, uterus anteroposterior diameter > 1 cm, uterine body-to-cervix ratio > 1, and ovarian volume ≥ 2 mL, regardless of patients’ age.

Basal hormone investigation is one of the first assessments in the suspicion of CPP, which is performed before the GnRH test. There are studies that have tried to simplify the diagnosis of CPP due to the lack of standardized cut-offs for gonadotropins after a stimulation test. For example, Neely et al., showed that baseline LH levels correlated with the LH peak and GnRH test results (r = 0.93, *p* < 0.0001), while baseline FSH was not useful for distinguishing children with CPP [5]. In another study, Cao et al., suggested that an elevated basal LH value (> 0.535 mIU/L as considered in their study, based on 1492 girls with CPP) could be used to diagnose CPP without a GnRH stimulation test [18]. The statistical analysis in our study confirmed these results and demonstrated the correlation between increased basal levels of FSH, LH, SHBG, and 17BE2 in patients with CPP. In addition to the LH peak, basal SHBG levels have also been shown to inversely correlate with BMI [19]. The results obtained from our cohort of patients with CPP did not show this correlation, although we observed that children with CPP had similar basal SHBG levels compared to children with a negative test.

Among the other parameters analyzed, we found no correlation with the lipid profile. Sørensen et al., studied the metabolic profile of 23 girls diagnosed with CPP compared to 115 children with normal pubertal development, finding that CPP girls had higher fasting insulin, triglycerides, and low-density lipoprotein (LDL) levels and also lower insulin sensitivity and lower HDL/cholesterol ratio [20]. In the follow-up, the authors also found that insulin sensitivity decreased during the first year of GnRH treatment, demonstrating that therapy did not improve the metabolic and cardiovascular risk profile of these girls [20]. Similar data were reported regarding the lipid profile and BMI, suggesting that CPP is a non-modifiable cardiovascular risk factor despite therapy [21]. Glucose metabolism is impaired in girls with CPP, as they have elevated levels of basal insulin [22], indicating a possible early onset of insulin resistance; this parameter may change or not following GnRH therapy, remaining high and even increasing in girls with a higher BMI [23].

The association between thyroid function and CPP is not fully understood, though it was first described by Jung et al., in a study involving 1247 patients (554 definitively diagnosed with CPP) who showed increased serum levels of TSH [24]. Age and LH surge were also shown to be independent predictors for serum TSH concentration in the same study [24]. Further confirmation of this relationship is the decrease in hyperthyrotropinemia after one year-therapy with GnRH agonists [25].

The onset of puberty is associated with a pubertal spurt and an increase in growth velocity, which leads to premature fusion of growth plates in the long run: growth velocity is considered pathological in subjects with suspected CPP when it rises to the upper percentile in an observation period of at least six months [4]. In our center, IGF-1 is not routinely measured in these patients; therefore, we were unable to investigate a potential correlation with the GnRH test. Serum levels of IGF-I and IGFBP-3 are known to be elevated during the pubertal spurt and have resulting increases in girls with CPP [26], while they tend to normalize in parallel with the reduction of growth velocity during treatment [27].

The retrospective design and the small population of children recruited for suspected CPP at the Fondazione Policlinico Universitario Agostino Gemelli IRCCS in Rome, Italy, are the main limitations of our single-center study, so these data are not generalizable to all patients with CPP. In addition, our cohort is mostly composed of females, which is consistent with the incidence rate of CPP. Because we did not have a sufficient number of males, we could not perform a subgroup analysis to evaluate differences based on sex. In addition, no association with diet or nutrition supplementation was assessed by our study.

## 5. Conclusive Remarks

Basal hormonal evaluations should be considered before a GnRH test and could help identify CPP. In particular, we found higher basal LH levels in children diagnosed with CPP (0.87 IU/L) compared to children without CPP (0.15 IU/L). Our analysis showed no significant association between pre-test auxological, metabolic, or radiological parameters and the final diagnosis of CPP. The multivariate analysis also showed that a higher LH peak (than that used in the guidelines) had excellent accuracy for the diagnosis of CPP (94% sensitivity and 95% specificity). Other parameters also showed high specificity and, in some cases, they might prefigure CPP diagnosis before performing the GnRH test. Further data involving a larger number of patients with CPP will help to verify these results in future prospective studies.

## Figures and Tables

**Figure 1 biomedicines-11-01678-f001:**
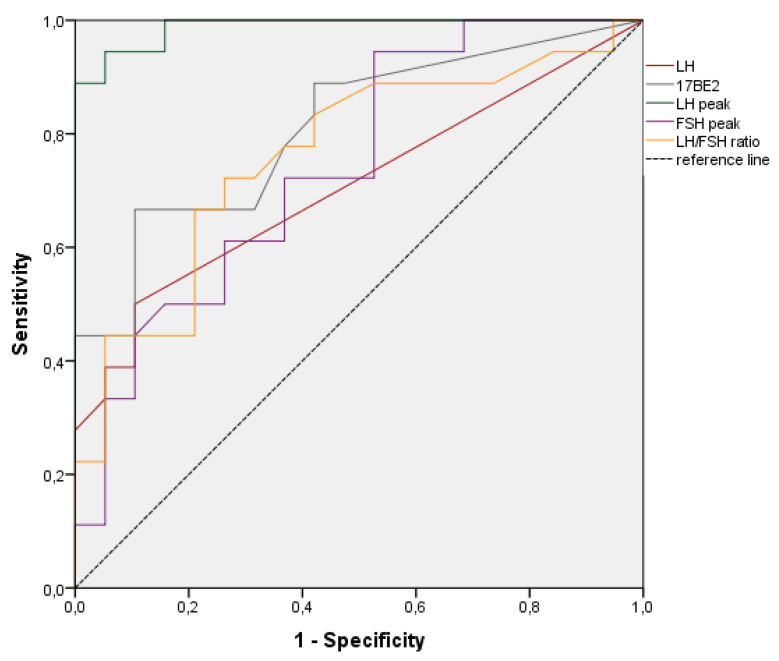
Receiver operating characteristic (ROC) curves for all variables significantly associated with central precocious puberty in the multivariate analysis.

**Table 1 biomedicines-11-01678-t001:** Characteristics of children with a clinical suspicion of central precocious puberty who were recruited in our study.

	Study Population (*N* = 37)
**Females**, n (%)	35 (94.6)
**Age** (years)	7.18 (2.38–7.72)
**Height** (cm)	124 (88.00–131.35)
**Z-score height**	0.5 (−1.33–1.18)
**Weight** (kg)	26.65 (15.67)
**Z-score weight**	0.42 (−1.41–1.34)
**BMI** (kg/m^2^)	16.89 (14.88–19.53)
**Z-score BMI**	0.3 (2.34)
**Adrenarche**, *n* (%)	5 (13.5)
**Familiar history of CPP**, *n* (%)	6 (16.2)
**TSH** (mU/L)	1.90 (1.35–4.41)
**fT3** (pg/mL)	3.82 (1.39)
**fT4** (ng/dL)	12.14 (1.68)
**Abnormal thyroid profile**, *n* (%)	10 (27.0)
**Glucose** (mg/dL)	76.59 (7.15)
**Hb** (g/dL)	12.99 (0.98)
**Platelet count** (×109/L)	317.51 (84.08)
**Abnormal blood count**, *n* (%)	6 (16.2)
**GOT** (IU/L) **Increased GOT**, *n* (%)	17 (14.00–19.50) 2 (5.4)
**Triglycerides** (mg/dL)	61 (48.5–73.5)
**Cholesterol** (mg/dL)	155.35 (25.13)
**LDL** (mg/dL)	89.11 (19.55)
**HDL** (mg/dL)	52.76 (13.40)
**Abnormal lipid profile**, *n* (%)	14 (37.8)
**FSH** (IU/mL)	3.4 (1.35–5.95)
**Increased baseline FSH**, *n* (%)	24 (64.9)
**LH** (mIU/mL)	0.1 (0.1–0.45)
**Increased baseline LH**, *n* (%)	5 (13.5)
**SHBG** (nmol/L)	87 (57.50–119.50)
**17BE2** (pg/mL)	19.00 (15.00–28.00)
**Prolactin** (ng/mL)	7.6 (5.3–12.6)
**Increased prolactin**, *n* (%)	13 (35.1)
**Increased bone age ≥ 1**, *n* (%)	17 (45.9)
**Pathological MRI ***, *n* (%)	3 (8.3)
**Uterus length ≥ 3.6 cm ****, *n* (%)	6 (17.1)
**Uterus antero–posterior diameter > 1 cm ****, *n* (%)	6 (17.1)
**Uterine body/cervix ratio > 1**, *n* (%)	8 (22.9)
**Ovarian volume ≥ 2 mL**, *n* (%)	15 (42.9)
**LH peak**	3.4 (1.4–8.4)
**FSH peak**	14.40 (8.55–23.45)
**LH/FSH ratio**	0.11 (0.007–0.25)
**LH/FSH ratio > 1**, *n* (%)	7 (18.9)
**LH/FSH ratio > 0.6**, *n* (%)	7 (18.9)
**CPP**, *n* (%)	18 (48.6)
**CPP etiology** #, *n* (%) ***Idiopathic*** ***Organic***	
	16 (88.9)
	2 (11.1)

Data are reported as mean (SD) or median (IQR) unless otherwise stated. * Not performed for *n* = 1. ** Not applicable for *n* = 2 (male patients). # Not applicable for *n* = 19 (patients without CPP). **Abbreviations:** BMI: body mass index, TSH: thyroid-stimulating hormone, fT3: triiodothyronine, fT4: thyroxine, Hb: hemoglobin, GOT: aspartate aminotransferase, LDL: low-density lipoprotein, HDL: high-density lipoprotein, FSH: follicle-stimulating hormone, LH: luteinizing hormone, SHBG: sex hormone binding globulin, 17BE2: 17-beta estradiol, MRI: magnetic resonance imaging, CPP: central precocious puberty.

**Table 2 biomedicines-11-01678-t002:** Comparison between children with and without central precocious puberty (PPC).

	Non-PPC Group (*N* = 19)	PPC Group (*N* = 18)	*p*
**Female***, n* (%)	17 (89.5)	18 (100)	0.49
**Age** (years), *median (IQR)*	7.32 (5.7–7.8)	7.02 (1.3–7.7)	0.37
**Height** (cm), *median (IQR)*	127.2 (110.0–133.0)	123.0 (76.5–130.3)	0.28
**Z-score height**, *median (IQR)*	0.5 (−2.1–1.3)	0.26 (−0.96–1.04)	0.73
**Weight** (kg), *median (IQR)*	30.0 (15.0–40.0)	24.5 (9.4–30.4)	0.13
**Z-score weight**, *median (IQR)*	1.05 (−1.1–1.81)	0.4 (−1.9–1.1)	0.27
**BMI**, *median (IQR)*	17.5 (13.5–23.3)	16.7 (15.1–18.2)	0.40
**Z-score BMI**, *mean (+/−SD)*	0.70 (+/−1.5)	−0.007 (+/−1.26)	0.13
**Adrenarche***, n* (%)	3 (15.8)	2 (11.1)	1
**Familiar history of CPP**, *n* (%)	2 (10.5)	4 (22.2)	0.40
**TSH**, *median (IQR)*	1.8 (1.6–5.4)	1.95 (1.3–3.1)	0.73
**fT3**, *mean (+/−SD)*	3.7 (+/−1.3)	3.98 (+/−1.5)	0.5
**fT4**, *mean (+/−SD)*	12.6 (+/−1.6)	11.6 (+/−1.6)	0.07
**Abnormal thyroid** **profile**, *n* (%)	5 (26.3)	5 (27.8)	1
**Glycemia** (mg/dl), *mean (+/−SD)*	78.2 (+/- 6.6)	74.9 (+/−7.5)	0.16
**Hb**, *median (IQR)*	12.9 (12.5–13.6)	13.0 (12.4–13.6)	0.62
**Platelet count**, *mean (+/−SD)*	324.8 (+/−91.5)	309.8 (+/−77.4)	0.59
**Abnormal blood cell count**, *n* (%)	4 (21.1)	2 (11.1)	0.66
**GOT**, *median (IQR)*	17.0 (14.0–20.0)	16.0 (13.7–19.0)	0.54
**Increased GOT***, n* (%)	1 (5.3)	1 (5.6)	1
**Triglycerides**, *median (IQR)*	61.0 (45.0–75.0)	60.0 (55.5–69.0)	0.66
**Cholesterol**, *mean (+/−SD)*	158.0 (+/−28.0)	152.5 (+/−22.2)	0.51
**LDL**, *mean (+/−SD)*	90.16 (+/−22.82)	88. 0 (+/−15.99)	0.74
**HDL**, *mean (+/−SD)*	53.11 (+/−14.11)	52.3 (+/−13.01)	0.87
**Abnormal lipid profile**, *n* (%)	10 (52.6)	4 (22.2)	0.09
**FSH**, *median (IQR)*	1.6 (1.0–3.4)	5.5 (3.1–8.2)	<0.001
**Increased baseline FSH**, *n* (%)	8 (42.1)	16 (88.9)	0.005
**LH**, *median (IQR)*	0.1 (0.1–0.1)	0.15 (0.1–1.5)	0.03
**Increased baseline LH**, *n* (%)	0 (0)	5 (27.8)	0.02
**SHBG**, *median (IQR)*	65.0 (42.0–99.0)	102.0 (75.2–128.5)	0.02
**17BE2**, *median (IQR)*	15.0 (15.0–20.0)	23.0 (18.7–44.2)	0.001
**Prolactin**, *median (IQR)*	8.0 (5.3–15.0)	7.5 (5.0–12.4)	0.68
**Increased prolactin**, *n* (%)	7 (36.8)	6 (33.3)	1
**Advanced bone age** **≥ 1 year**, *n* (%)	12 (63.2)	8 (44.4)	0.25
**Pathological brain MRI**, *n* (%)	1 (5.3)	2 (11.8)	0.59
**Uterus length ≥ 3.6 cm ****, *n* (%)	2 (11.8)	4 (22.2)	0.66
**Uterus antero–posterior** **diameter > 1 cm *****, n* (%)	2 (11.8)	4 (22.2)	0.66
**Uterine body/cervix ratio** **> 1**, *n* (%)	3 (17.6)	5 (27.8)	0.69
**Ovarian volume ≥ 2 mL**, *n* (%)	7 (41.2)	8 (44.4)	0.84
**LH peak**, *median (IQR)*	1.4 (0.7–2.2)	8.4 (6.7–26.8)	<0.001
**FSH peak**, *median (IQR)*	10.5 (5.1–19.0)	18.9 (10.2–44.6)	0.01
**LH/FSH ratio**, *median (IQR)*	0.008 (0.005–0.048)	0.109 (0.01–0.76)	0.006
**LH/FSH ratio** > 1, *n* (%)	0 (0)	7 (38.9)	0.003

** Not applicable for *n* = 2 (male patients). **Abbreviations:** BMI: body mass index, TSH: thyroid-stimulating hormone, fT3: triiodothyronine, fT4: thyroxine, Hb: hemoglobin, GOT: aspartate aminotransferase, LDL: low-density lipoprotein, HDL: high-density lipoprotein, FSH: follicle-stimulating hormone, LH: luteinizing hormone, SHBG: sex hormone binding globulin, 17BE2: 17-beta estradiol, MRI: magnetic resonance imaging.

**Table 3 biomedicines-11-01678-t003:** Multivariate linear regression adjusted for sex and age including all factors with *p* < 0.1 in the univariate analysis.

Variable	F-Test	*p*
fT4	2.247	0.101
Abnormal lipid profile	1.590	0.210
FSH	0.952	0.427
LH	4.354	0.011
SHBG	1.166	0.337
17BE2	4.968	0.006
LH peak	5.425	0.004
FSH peak	23.457	<0.001
LH/FSH ratio	4.801	0.007

**Abbreviations:** fT4: thyroxine T4, FSH: follicle-stimulating hormone, LH: luteinizing hormone, SHBG: sex hormone binding globulin, 17BE2: 17-beta estradiol.

**Table 4 biomedicines-11-01678-t004:** Accuracy with 95% confidence intervals, *p*-values, best cut-off values, sensibility, and specificity for each variable associated with the diagnosis of central precocious puberty in the multivariate analysis.

Variable	Accuracy	95%CI	*p*	Best Cut-off	Sensitivity (%)	Specificity (%)
LH	0.708	0.536−0.879	0.031	0.15	50	90
17BE2	0.811	0.671–0.952	0.001	21.5	67	90
LH peak	0.988	0.964–1.000	0.001	4.10	94	95
FSH peak	0.744	0.586–0.902	0.010	21.5	44	90
LH/FSH	0.762	0.603–0.907	0.007	0.023	72	74

**Abbreviations:** LH: luteinizing hormone, 17BE2: 17-beta estradiol, FSH: follicle-stimulating hormone.

## Data Availability

No datasets were generated and analyzed during the study.

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
