# Peer review of "Auxological and Metabolic Parameters of Children Undergoing the Gonadotropin-Releasing Hormone Stimulation Test: Correlations with the Final Diagnosis of Central Precocious Puberty in a Single-Center Study"

_biomedicines, 2023, doi:10.3390/biomedicines11061678_

Round 1

Reviewer 1 Report

I consider that this study is conducted and described generally well besides the following major and minor points.

Major comment

Figure 1 is not properly depicting the HPG axis especially the brain and the pituitary. "Hypothalamus" should be shown. Where are the GnRH cells? Indicate the "Anterior pituitary". What are the molecules like things shown as LH and FSH? Show the real three dimensional structures of the molecules if possible.

Lines 169-179: Statistical analysis should be moved to the method.

Minor comment

Line 61: "kisspeptin".

Lines 67-70: I could not get the meaning of this sentence.

Author Response

REVIEWER 1

I consider that this study is conducted and described generally well besides the following major and minor points.

Dear Reviewer, thank you so much for your comments on our paper, which helped us to improve it.

Major comment: Figure 1 is not properly depicting the HPG axis especially the brain and the pituitary. "Hypothalamus" should be shown. Where are the GnRH cells? Indicate the "Anterior pituitary". What are the molecules like things shown as LH and FSH? Show the real three dimensional structures of the molecules if possible.

Thank you for your comment. We chose to delete Figure 1 as it was deemed unnecessary by reviewer 3. We have inserted a new figure in the paper, deriving from the multivariate statistical analysis, which has been added.

Lines 169-179: Statistical analysis should be moved to the method.

Done. We moved the paragraph to the Patients and Methods section. We also implemented our statistical analysis with a multivariate analysis (which includes the ROC curves).

Minor comment: Line 61: "kisspeptin".

Modified, thank you.

Lines 67-70: I could not get the meaning of this sentence.

We have changed that sentence to make it clearer. We wanted simply to highlight that LH value can prefigure the diagnosis of CPP. Thanks again for your comment.

Reviewer 2 Report

I found the principle aim of this study rather confusing, which was to establish  potential correlation between  auxological, metabolic and radiological parameters in a cohort 72 patients with  the final diagnosis of CPP compared to those who did not have this  diagnosis.  Ideally the endocrinologist needs to know in which patients to do a GnRH test - and which patient there is no need for the test as the diagnosis is either already confirmed or can be confidently refuted. I don't think a significant difference between the mean value of relevant parameters is enough evidence that this approach would be clinically useful , although if there is no difference between parameters then the utility of the marker is clearly questionable. 

I think as a minimum ROC analysis of the data is required.  This would establish accuracy, and a sensitive and specific cut-off for each parameter chosen.  However there are already more sophisticated and larger mulitparameter  analsyses published on this topic doi: 10.3390/diagnostics13091550    and those cited in the discussion. 

Author Response

REVIEWER 2

I found the principle aim of this study rather confusing, which was to establish potential correlation between auxological, metabolic and radiological parameters in a cohort of 72 patients with the final diagnosis of CPP compared to those who did not have this diagnosis. Ideally the endocrinologist needs to know in which patients to do a GnRH test - and which patient there is no need for the test as the diagnosis is either already confirmed or can be confidently refuted. I don't think a significant difference between the mean value of relevant parameters is enough evidence that this approach would be clinically useful, although if there is no difference between parameters then the utility of the marker is clearly questionable. 

Dear Reviewer, thank you for your useful criticism on our paper, which helped us to improve it. As mentioned in the limitations of our study, ours is a single-centre study, with a limited sample size and, therefore, results of our statistical analysis may not be generalizable to the general pediatric population. Despite this, we found the results of our analysis interesting, coming from a single Italian tertiary hospital. We have made changes in various sections of our article, including a multivariate statistical analysis, and we hope that these changes will satisfy your requests.

I think as a minimum ROC analysis of the data is required. This would establish accuracy, and a sensitive and specific cut-off for each parameter chosen. However there are already more sophisticated and larger mulitparameter analsyses published on this topic (doi: 10.3390/diagnostics13091550) and those cited in the discussion. 

Thanks for your suggestion. We have modified the statistical analysis section, which is now located in the Patients and Methods section, as required by reviewer 1. We have implemented our work with a multivariate analysis. ROC analysis, as required, has been included in the paper.

Reviewer 3 Report

This manuscript is intended to provide useful information about the use of measurements, lab tests, and imaging to distinguish between patients with hormonally confirmed CPP vs those children with a negative GnRH test.  This area has been the subject of numerous publications over the years and it is not clear that this small study involving 37 girls adds anything to the literature.  Specific comments

1.  Figure 1 can be deleted.  readers should be very familiar with the HPG axis.

2. Table 1:  The Z score of BMI is given as 3.02 which cannot be correct. Did they mean 0.3?  

3.  What do M-stadium and P-stadium mean?  They are not standard abbreviations.

4. Table 1 has a lot of data which should be omitted.  Why include CBC parameters?  Is it really needed to include thyroid testing and prolactin which are not known to be abnormal in CPP?  And what is GTX?

5. Section 2.3 line 4:  should read "subsequent serial SAMPLING of LH and FSH.  Line 5: please delete line about "to facilitate blood sampling..." which requires no explanation

6. Section 2.3 paragraph 2:  the section about how administration of GnRH is generally safe can be deleted, or if necessary moved to the discussion

7. Section 3.2:  what type of CNS tumor was found?  Also, empty sella may be associated with GH deficiency but should not be considered a pathologic finding for CPP.

8. Most centers use an LH/FSH ratio of >0.6 as the cut-off for CPP and the authors should reanalyze their data using this cut-off.

9.  Table 2 has many problems.  First, it is puzzling that the authors found no differences between the CPP and non-CPP groups in height Z-score, bone age advance, uterine size (they need to define body/neck relationship) and ovarian volume as others have found, which makes one wonder how distinct their 2 groups really are.  They give mean basal LH as 0.15 whereas in Fig 2 it looks closer to 0.5, and only 28% had increased basal LH, in contrast to most other studies.  Also basal FSH is stated as 5.5 whereas Figure 2 shows it as closer to 10.  The mean LH/FSH ratio is stated as 0.109 in the CPP group whereas based on the mean peak LH and FSH it should be in the 0.4-0.5 range.  Also it is puzzling that in this study the basal FSH is high in 88% whereas other studies do not find the basal FSH to be diagnostically useful

Author Response

REVIEWER 3

This manuscript is intended to provide useful information about the use of measurements, lab tests, and imaging to distinguish between patients with hormonally confirmed CPP vs those children with a negative GnRH test. This area has been the subject of numerous publications over the years and it is not clear that this small study involving 37 girls adds anything to the literature. 

Dear Reviewer, thank you for your criticism and comments on our paper. As mentioned in the limitations of our study, this is a single-centre study, with a limited sample size and, therefore, the results of our experience with its resulting statistical analysis may not be generalizable.

Specific comments: Figure 1 can be deleted. Readers should be very familiar with the HPG axis.

Thanks for your comment. Figure 1 was specifically requested by the editor during the first article submission process; Reviewer 1 asked us for changes to make it more specific. We have chosen to eliminate it, also because of your suggestion. We have now inserted another figure from the multivariate statistical analysis that we have added to our paper, following Reviewer 2’s comments.

  1. Table 1: The Z score of BMI is given as 3.02 which cannot be correct. Did they mean 0.3?  

Thanks for your comment. We changed the value, it was an error during the copyediting of the article.

  1. What do M-stadium and P-stadium mean?  They are not standard abbreviations.

Thanks for your remark. These are the abbreviations used to indicate mammary and pubertal development. We have removed this information from the Table to make it more intuitive.

  1. Table 1 has a lot of data which should be omitted. Why include CBC parameters?  Is it really needed to include thyroid testing and prolactin, which are not known to be abnormal in CPP?  And what is GTX?

Thank you for your comment. As stated by the Italian national guidelines (see reference 4 - http://www.siedp.it/files/PDTAPubertprecocecentrale_approvato.pdf) basal blood exams are always recommended, in the suspicion of CPP, despite the parameters you mentioned are often normal (in patients with a positive GnRh test). This also applies to thyroid function, but this is indicated by SIEDP: "...among the laboratory criteria, the study of thyroid function must always be performed in patients with suspected precocious puberty, especially in females with precocious thelarca…".

We were unable to confirm any statistically significant difference between these parameters, but we thought it was also interesting to analyze data in our retrospective evaluation of patients’ medical charts. We have made adjustments to methods, table 1 and results section. We thank you again for your request of precisation.

  1. Section 2.3 line 4: Should read "subsequent serial SAMPLING of LH and FSH.  Line 5: please delete line about "to facilitate blood sampling..." which requires no explanation.

Modified, thank you.

  1. Section 2.3 paragraph 2: the section about how administration of GnRH is generally safe can be deleted, or if necessary moved to the discussion.

Modified, thank you.

  1. Section 3.2: what type of CNS tumor was found? Also, empty sella may be associated with GH deficiency, but should not be considered a pathologic finding for CPP.

Thank you for your comment, we have answered to these questions in the proper section.

  1. Most centers use an LH/FSH ratio of >0.6 as the cut-off for CPP and the authors should reanalyze their data using this cut-off.

Thanks for your comment. Also in this case we used the Italian cut-off (>1 as suggested by our guidelines). It is also absolutely interesting to analyze the cut-off of 0.6 provided by many centers and other international guidelines, therefore we have integrated this information into the text. We also modified the results of the statistical analysis, integrating the results of the multivariate analysis and ROC analysis.

  1. Table 2 has many problems. First, it is puzzling that the authors found no differences between the CPP and non-CPP groups in height Z-score, bone age advance, uterine size (they need to define body/neck relationship) and ovarian volume as others have found, which makes one wonder how distinct their 2 groups really are. They give mean basal LH as 0.15 whereas in Fig 2 it looks closer to 0.5, and only 28% had increased basal LH, in contrast to most other studies.  Also basal FSH is stated as 5.5 whereas Figure 2 shows it as closer to 10. The mean LH/FSH ratio is stated as 0.109 in the CPP group whereas based on the mean peak LH and FSH it should be in the 0.4-0.5 range. Also it is puzzling that in this study the basal FSH is high in 88%, whereas other studies do not find the basal FSH to be diagnostically useful.

Thank you for your comments on our table, which has been clarified. Conversely, figure 2 has been removed. We have created another figure, based on the results of our novel statistical analysis. We hope these changes may satisfy the need of clarity. Regarding the other parameters, we are not surprised by such results, especially for the information coming from the pelvic ultrasound: there is no univocal consensus for the diagnosis of CPP and pelvic ultrasound assessments, although they should provide important clues. It should also be specified how we considered these parameters to define them as "increased": uterus length ≥ 3.6 cm, AP diameter > 1 cm, ratio >1, ovarian volume ≥ 2 ml. These cut-offs are reported in our guidelines (ref. n 4) and are unique for all prepubertal children (regardless of age). To date, we do not have standardized percentiles for gynecological measurements in Italian children. We specified all these considerations in the main text. Thank you again.

Round 2

Reviewer 1 Report

I consider that the authors have adequately revised the manuscript.

Author Response

Thank you so much for appreciating the revision of our paper

Reviewer 2 Report

The authors have addressed all of my suggestions, thankyou

Author Response

Thank you so much for appreciating all our efforts related to the revision of this paper

Reviewer 3 Report

I still have reservations about the significance of the results, given the large number of studies which looked at the same predictors, but the authors have addressed many of my concerns.  

I did find one clear error in Table 4.  The cut-off for LH/FSH is given as 0.023, but that cannot be correct.

Author Response

We all thank the reviewer 3 for his valuable advice, which allowed us to significantly improve our study.

Thank you also for giving us the opportunity of revising the accuracy of Table 4: after reviewing the statistical analysis, we confirm that the best cut-off value for the LH/TSH ratio is 0.023.

We understand that this cut-off value for the LH/FSH ratio may seem incorrect compared to other studies where it is higher (for example, 0.07 in Cao R, et al. The Diagnostic Utility of the Basal Luteinizing Hormone Level and Single 60-Minute Post GnRH Agonist Stimulation Test for Idiopathic Central Precocious Puberty in Girls. Front Endocrinol 2021;12:713880.

However, in our sample of patients, the LH/FSH ratio in subjects without CPP had a median value of 0.008 (0.005-0.048), as described in Table 2, whereas it had a median value of 0.109 (0.01-0.76) in patients with CPP.

In addition, several controls had an LH/FSH ratio of 0.01 or less.

In view of these data, the result of our statistical analysis regarding the cut-off value of the LH/FSH ratio should no longer seem incorrect.

Thank you again for appreciating all our efforts, related to this paper, dealing with our single-centre experience in children with central precocious puberty.